# NFV/SDN as an Enabler for Dynamic Placement Method of mmWave Embedded UAV Access Base Stations

**Gia Khanh Tran \*** , **Masanori Ozasa and Jin Nakazato**

Department of Electrical and Electronic, Tokyo Institute of Technology, Tokyo 152-8550, Japan
\* Correspondence: khanhtg@mobile.ee.titech.ac.jp

**Abstract:** In the event of a major disaster, base stations in the disaster area will cease to function, making it impossible to obtain life-saving information. Therefore, it is necessary to provide a wireless communication infrastructure as soon as possible. To cope with this situation, we focus on NFV/SDN (Network Function Virtualization/Software-Defined Networking)-enabled UAVs equipped with a wireless communication infrastructure to provide services. The access link between the UAV and the user is assumed to be equipped with a millimeter-wave interface to achieve high throughput. However, the use of millimeter-waves increases the effect of attenuation, making the deployment of UAVs problematic. In addition, if multiple UAVs are deployed in a limited frequency band, co-channel interference will occur between the UAVs, resulting in a decrease in the data rate. Therefore, in this paper, we propose a method that combines UAV placement and frequency division for a non-uniform user distribution in an environment with multiple UAVs. As a result, it is found that the offered data rate is improved by using our specific placement method, in terms of not only the average but also the outage user rate.

**Keywords:** UAV base station; mmWave; K-means method; fractional frequency reuse; NFV/SDN

---





## 1. Introduction

In recent years, the total traffic in mobile networks has been increasing at an average annual rate of 46% due to the explosive growth of various applications such as video streaming services [1]. To cope with this problem, 5G (5th Generation Mobile Communication System) services were launched worldwide in 2019. In this 5G era, not only human beings but also objects have been connected to the Internet as a social infrastructure. This trend is expected to continue in the future, with the mobile communication system expected to function as a fundamental infrastructure of society in the Beyond 5G/6G era [2]. As a result, it has become more important than ever to be "connected" in any place and under any circumstances. For this reason, UAVs (Unmanned Aerial Vehicles) have, in recent years, become a familiar part of our lives and are expected to be utilized in various fields such as infrastructure monitoring, delivery, and disaster investigation. In this paper, we focus on the use of access UAVs as temporary base stations for deploying wireless networks (UAV networks) [3], knowing that our investigation on backhaul UAVs was studied in [4,5].

UAV networks have several advantages because of their ability to place base stations in the sky. The first is that they can move regardless of the constraints on the ground. This allows for rapid deployment of base stations when needed and allows for optimal placement of base stations for a particular distribution of users. The other point is the ability to provide data from the sky. This reduces the probability of being blocked by buildings, etc., and increases the probability that the propagation path between the UAV and the user is in line-of-sight condition. These advantages make UAV networks attractive applications for many use cases but, in this study, we mainly consider disaster-stricken areas as our target use case. For example, the Great East Japan Earthquake in 2011 caused great damage to the Pacific coast of Japan. As a result of the earthquake, about 29,000 cell phone

base stations and PHS (Personal Handy-phone System) base stations were shut down [6], disabling the communication of life-threatening information such as safety information and evacuation information to/from cell phone terminals. In such situations, it necessitates the establishment of a temporary wireless network that can be constructed quickly and stably. To this end, these requirements can be realized by exploiting the mobility of UAVs.

The placement of access UAVs in UAV networks has been studied from various viewpoints. In [7], a method to place UAVs to cover users of degraded data rate located at the edge of the service area was studied. In [8], the placement was optimized to reduce the total transmission power of UAVs while ensuring service requirement constraints of each user. In [9], the placement of UAVs was obtained by solving a packing problem on the service area to enhance the system's coverage. In [10], UAV placement was further optimized by solving a particle swarm optimization problem to maximize the area coverage. In these conventional studies, the microwave bands of limited bandwidth were assumed since these lower frequency bands are expected to attain higher connectivity. However, for future applications including digital twins to support rescue activities using remotely controlled equipment in disaster areas, higher data rate transmission will be become indispensable. Furthermore, the data volume of web browsing is increasing yearly along with the increase of communication speed [11], so large-capacity transmission will also be necessitated for information collection in disaster areas [4]. The bandwidth of the microwave band is not sufficient to support such use cases. In other words, it necessitates the introduction of higher frequency bands such as millimeter-wave to guarantee a larger system bandwidth [12]. Furthermore, the miniaturization of devices is an important issue for UAVs, and the use of frequencies with shorter wavelengths such as millimeter-wave is more advantageous in this respect. Regardless of several disadvantages when using these higher frequency bands, e.g., limited coverage, such issues have been overcome by introducing a high-directivity beamforming technique together with a fast-beam switching mechanism. For these reasons, nowadays, millimeter-wave is even being introduced into 5G cellular networks [13] and its employment for UAV applications is also being discussed by 3GPP and other organizations [14].

Different from conventional works, this paper therefore studies the placement of access UAV networks employing broadband millimeter-wave. As mentioned above, in the millimeter-wave band, the effects of distance attenuation and blocking are significant, so UAV placement concerning the user distribution becomes an important issue. However, there has been little study of the deployment of UAV networks in the millimeter-wave band. The following is a partial list of references that discussed UAV deployment in millimeter-wave bands. In [15], the goal was to maximize the number of users that can be covered, with convolutional neural networks used to increase computational efficiency. In [16], an algorithm to solve the placement problem to achieve the required SINR (Signal-to-Interference-and-Noise Ratio) for all users was considered. In [17], the placement problem of UAVs was considered to maximize the total data rate provided to the user. In this paper, we consider the placement problem with a different objective from these references, i.e., to improve the overall user rate's cumulative distribution function (CDF), as will be shown in Section 4. In addition, in the conventional literature, the distribution of users is assumed to be uniform. However, in the real environment of mobile communication, users are not uniformly distributed, but rather non-uniformly distributed. Since the maximum number of users accommodated by a single base station is limited to attain a target desired user rate, whether the user distribution is uniform or biasedly non-uniform is a significant factor when considering the problem of UAV base station placement. For these reasons, in this paper, we propose a new approach to non-uniform user distribution in the millimeter-wave band. Specifically, the placement of millimeter-wave band UAV networks with various user distributions will be investigated.

In summary, this paper deals with the problem of deploying access UAVs to increase the downlink data rate provided to stationary users distributed non-uniformly on the ground. UAV base stations are dynamically deployed, unlike ground base stations, and

the spacing between UAV base stations is irregular, so there can be various situations of coverage overlap. Therefore, interference from other UAV base stations is a major problem. For this reason, after studying the placement method, this paper furthermore introduces an interference reduction method called Fractional Frequency Reuse (FFR) to overcome the severe interference issue. Various control protocols for UAVs for different purposes including security/surveillance have been studied by other researchers, e.g., [18] is thus outside the scope of this paper. In this paper, we will focus only on the communication network aspect rather than control protocols of UAVs. Numerical analyses are conducted in Section 4 to confirm the effectiveness of our proposed method, in terms of not only average user rate but outage user rate. This paper extends from the authors' previous work in [19], where additional comparison results against a conventional microwave-band system are included in Section 4. Our numerical results further reveal that the proposed method's performance in the millimeter-wave band is superior to that in the conventional microwave band. Thus, this paper makes an unparalleled academic contribution to the field. Moreover, the paper will also explain how to construct the proposed network using NFV/SDN technologies as enablers.

This paper is organized as follows. First, Section 2 presents an overview of how NFV/SDN technologies are suitable for the dynamic deployment of the UAV network by referring to our conventional work as well as the recent literature. Section 3 describes the architecture and system model in this study, Section 4 explains the proposed method, Section 5 presents the numerical results, and, finally, Section 6 concludes the paper.

## 2. NFV/SDN as an Enabler for UAV Network-Related Work

NFV (Network Function Virtualization) and SDN (Software Defined Network) are needed as key technologies to achieve variable control of networks and computing for UAV networks. Here, NFV is a platform for deploying applications with computing resources, which is known as MEC (Multi-Access Edge Computing). In [20], the authors proposed an algorithm for controlling backhaul between network devices using OVS (Open Virtual Switch) and Open Flow-based SDN under varying user requirements and showed its effectiveness through numerical analysis. In [21], a testbed was constructed to demonstrate the effectiveness of the proposed algorithm in tracking applications and networks as users move. Specifically, the SDN controller detects user context information, migrates applications to the user's destination using MEC-NFV, and controls the backhaul network with SDN. On the other hand, the system was extended to millimeter-wave V2X (Vehicle-to-Everything) scenarios and evaluated through field demonstration experiments [22]. SDN controls the backhaul to obtain other RSUs (Road Side Units), enabling cooperative communication, which supports safer automated driving.

Furthermore, we have proposed an algorithm that dynamically controls not only the backhaul network but also the power supply of the base station by SDN according to the traffic conditions of the user [23]. The proposed method effectively satisfies both user satisfaction and power supply connectivity. Furthermore, the significance of using NFV technology to run applications on MEC connected by network control was shown from the perspective of the operator holding the MEC [24]. In addition, the NFV/SDN controller was designed from an E2E viewpoint for MEC, considering both the application and the network [25]. PoC (Proof-of-Concept) of [25] validated the effectiveness of the MEC in the outdoor field.

Regarding UAV networks, various studies have discussed [26–28] taking advantage of SDN. As mentioned above, SDN can be applied to UAVs to maintain a backhaul network and connect to nearby base stations in a dynamically changing UAV network configuration. Moreover, by adding MEC-NFV as well as SDN in [29–31], it is possible to deploy simple applications (e.g., cache contents) by holding a small computing resource in the UAV.

From a different aspect, signaling control with NFV/SDN was discussed in [32–34]. In [32], an algorithm was proposed for finding user pairing and power allocation that can maximize the wireless communication capacity using the NOMA (Non-Orthogonal

Multiple Access) at the access side. In [33], an interference control method was proposed by changing the number of UAVs and their deployment method to an objective function and flexibly changing them to take advantage of SDN features. In contrast to the above two references, this paper proposes a frequency division method focusing on the access side and thus has a different objective target. On the other hand, [34] proposed an algorithm to control the traffic signal to reduce the energy of computation and latency on the UAV side. Different from [34], this paper proposes a control method that can maximize throughput on the UAV side.

For these reasons, this paper employs NFV/SDN as a key enabler for a dynamic construction of our proposed UAV networks.

## 3. Architecture and System Models

In this section, we describe the architecture and system model used in this study.

### 3.1. Architecture

Figure 1 shows the overall architecture [4]. UAVs can be placed in the sky without being affected by ground restrictions, thus ensuring a higher probability of a line-of-sight environment, which is important in the millimeter-wave band. In addition, since the base station itself is mobile, it is possible to quickly construct a network in a disaster area. Moreover, for the selection of the types of UAVs, this study assumes the use of a multirotor UAV rather than a fixed-wing UAV for our temporary base stations. Unlike fixed-wing UAVs, multirotor UAVs have a higher degree of freedom of movement, enabling the optimization of not only the placement but also the UAV's trajectories [35].

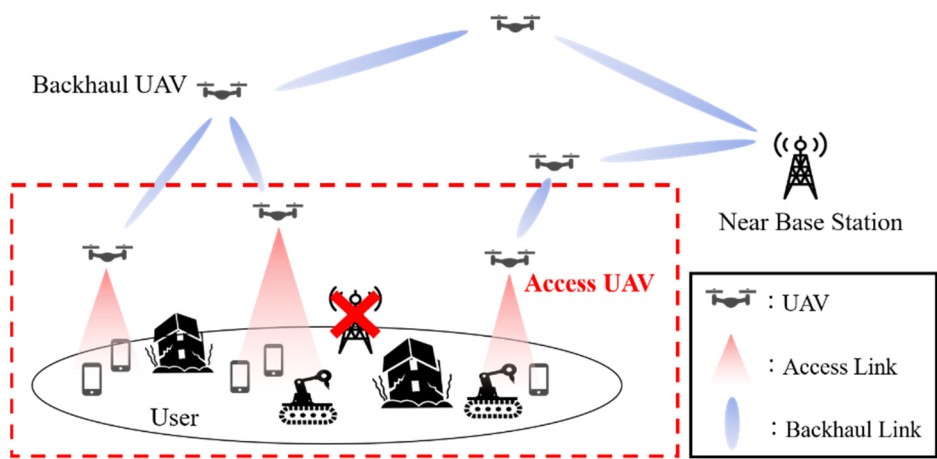

**Figure 1.** System architecture.

As shown in Figure 1, the system consists of two types of UAVs: access UAVs and backhaul UAVs. The backhaul UAV is responsible for relaying the traffic sent from the base station on the ground and other backhaul UAVs to other UAVs. If the distance between the access UAV and the neighboring base station is fixed, by using backhaul UAVs as relays in between base stations and access UAVs, the transmission distances between these links are effectively shortened. The shorter communication distance between UAVs alleviates the effects of distance attenuation and rain attenuation, which are problems in the millimeter-wave band. This allows for a longer communication distance between the access UAV and the ground base station, which can be used for various use cases. The access UAV provides the traffic sent from the backhaul UAV to the user on the ground. Since the access UAV provides data directly to the user, its placement has a significant impact on the data rate provided, which is the main focus of this paper.

*3.2. System Models*

As described in the previous section, this paper examines access UAVs. The backhaul [4,5] is assumed to be ideal for a separate study. When the entire system is considered, the joint optimization of both backhaul and access UAVs against a fixed user distribution might have a significant impact on the overall system's performance [36]. However, due to the moving freedom of the UAVs, we can properly optimize the backhaul UAV placement to fulfill the required communication rates of the fixed access UAVs, as conducted in our previous work [4]. For this reason, for ease of investigation in this study, we separated the discussion of the backhaul UAV placement from that of the access UAV placement problem, such that the placement of the latter UAV will merely depend on the provided user distribution. The joint optimization of both access and backhaul UAVs will be investigated in our future work.

In this paper, we assume that users always request data (full-buffer model). It is also assumed that the user distribution is known. In recent years, there has been much research on estimating user distributions, e.g., [37], so the estimation of user distributions is outside the scope of this paper. Therefore, we do not limit ourselves to a user's specific application but aim to improve the offered data rate in each situation.

3.2.1. Air-to-Ground Path Loss Model

The path loss model for the UAV in the sky and the user on the ground in this paper follows the model in [38]. In [38], the air-to-ground propagation path is first divided into the probability of a Line of Sight (LoS) environment and the probability of a Non-Line of Sight (NLoS) environment. The path loss, in this case, is as follows:

$$L = P_{\text{LoS}} L_{\text{LoS}} + P_{\text{NLoS}} L_{\text{NLoS}} \tag{1}$$

where $L$ is the propagation loss, $P_{\text{LoS}}$ and $P_{\text{NLoS}}$ are the probabilities that the propagation path becomes LoS or NLoS, respectively, and $L_{\text{LoS}}$ and $L_{\text{NLoS}}$ are the corresponding propagation losses for these two cases. The probability that the propagation path becomes LoS is determined according to the environment of the corresponding area as follows:

$$P_{\text{LoS}} = \frac{1}{1 + a \exp(-b[\theta - a])} \tag{2}$$

where $a$ and $b$ are constants determined by the environmental parameters of the relevant area [38] and $\theta$ is the elevation angle from the user on the ground to the UAV. ($P_{\text{LoS}}$ is a function based on the user's position, and the total integration of $P_{\text{LoS}}$ over all solid angles does not necessarily yield a unit value.)

Since the propagation path can be classified into two types, LoS and NLoS, the NLoS probability is as follows:

$$P_{\text{NLoS}} = 1 - P_{\text{LoS}} \tag{3}$$

The path loss of LoS and NLoS can be divided into propagation loss and additional loss ($\eta$) when the space is free, which can be expressed as follows:

$$L_{\text{LoS/NLoS}} \, [\text{dB}] = L_{\text{free}} + \eta_{\text{LoS/NLoS}} \tag{4}$$

where $\eta$ represents the additional loss in LoS or NLoS condition, respectively, which is determined by the frequency band and is described in Section 4. $L_{\text{free}}$ [dB] represents the free-space propagation loss, which can be expressed as follows:

$$L_{\text{free}} \, [\text{dB}] = 20 \log d + 20 \log f + 20 \log \left( \frac{4\pi}{c} \right) \tag{5}$$

where $d$ [m] is the linear distance between the UAV and the user, $f$ [Hz] is the carrier frequency, and $c$ [m/s] is the speed of light.

Substituting Equations (2)–(5) into Equation (1), we finally obtain the propagation loss as follows:

$$L\,[\text{dB}] = \frac{\eta_{\text{LoS}} - \eta_{\text{NLoS}}}{1 + a\,\exp\left(-b\left[\arctan\left(\frac{h}{R}\right) - a\right]\right)} + 10\log\left(h^2 + R^2\right) + 20\log f + 20\log\left(\frac{4\pi}{c}\right) + \eta_{\text{NLoS}} \tag{6}$$

where $h\,[m]$ is the altitude of the UAV and $R\,[m]$ is the horizontal distance between the UAV and the user, respectively. These are also shown in Figure 2.

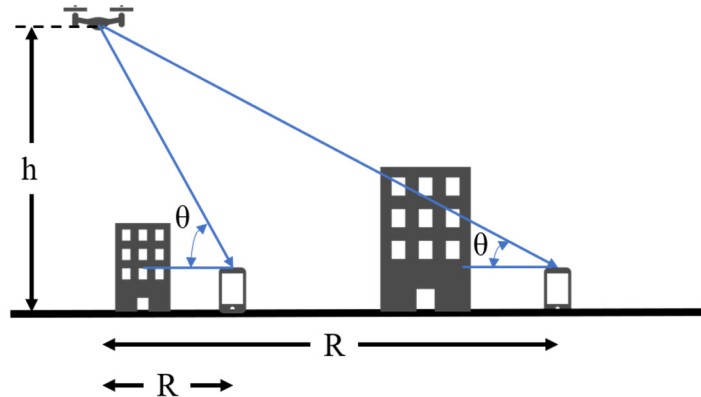

**Figure 2.** Parameters related to positioning.

### 3.2.2. Calculation Model for Communication Capacity

Using the Friis transmission equation [39], the received power is calculated in dB notation as follows:

$$P_{\text{r}}\,[\text{dBW}] = P_{\text{t}} + G_{\text{t}} + G_{\text{r}} - L \tag{7}$$

where $P_{\text{t}}\,[\text{dBW}]$ and $P_{\text{r}}\,[\text{dBW}]$ denote transmit and receive power, respectively. $G_{\text{t}}\,[\text{dB}]$ and $G_{\text{r}}\,[\text{dB}]$ are the gain of the transmit and receive antennas, respectively; knowing that when we specify a parameter/variable with the term [dB], the value is computed in a logarithmic scale rather than in a linear scale. Using the received power obtained here, the SINR is calculated as follows:

$$\gamma = \frac{P_{\text{r}}}{P_{\text{n}} + I} \tag{8}$$

where $\gamma$ and $I$ [W] are the SINR and interference power, respectively, and $P_{\text{n}}$ [W] is the thermal noise power, which is calculated as follows:

$$P_{\text{n}} = k_{\text{B}}\,T\,B \tag{9}$$

where $k_{\text{B}}$ [J/K] is Boltzmann's constant, $T$ [K] is the absolute temperature, and $B$ [Hz] is the bandwidth of the system.

The communication capacity is calculated by the following equation, based on Shannon's communication capacity theorem [40], which indicates the marginal capacity.

$$C\,[\text{bps}] = B\log_2(1 + \gamma) \tag{10}$$

Since this paper assumes a TDMA (Time Division Multiple Access) communication method, the communication capacity per single user is obtained by dividing the communication capacity per UAV by the number of users accommodated by the UAV.

## 4. Proposed Method

In this section, a snapshot of the instantaneous user distribution is assumed. First, we propose a placement method for UAVs after the user distribution is obtained, then we propose a frequency division method after the placement decision is made. When

considering this problem, the data rate can be improved by increasing the number of UAVs. However, this has drawbacks in terms of the overall deployment cost and power consumption of the system. For this reason, in this paper, we only consider a scenario with a limited number of UAVs, ranging from 3 to 10. A maximum number of 10 was selected so that UAVs are separated by certain distances to prevent them from colliding with each other in our evaluation area of interest. We would like to investigate the optimal number of UAVs under interference constraints in our future work.

In this paper, we use the method described in [38] to determine the altitude of the UAV. This method determines the altitude of each UAV coverage circle in such a way that the propagation loss of the user at the edge of the circle is minimized. Therefore, we now describe the proposed method for the horizontal placement of UAVs.

*4.1. K-Means Algorithm + Smallest-Circle Problem*

As mentioned above, we use the millimeter-wave band for the access link in this study. Since the millimeter-wave band has a higher frequency than the microwave band used so far, the propagation loss is larger, as shown in Equation (5). To improve the data rate provided to the user, it is necessary to reduce the propagation loss, with Equation (5) showing that it is necessary to shorten the distance between the user and the UAV.

As a previous method to shorten the distance between user and UAV, [41] uses the K-means method to derive the horizontal position of the UAV. The K-means method is a clustering method, in which a set of points is divided into multiple sets, minimizing as much as possible the sum of the Euclidean distances between the centers of the clusters and the points in each cluster.

In [41], the user distribution is considered as a point cloud, while the resulting clusters are the set of users covered by each UAV and the cluster center, which is the center of gravity of each cluster, is the horizontal position of the UAV.

The authors of [38] use another comparison scheme; however, this method is only effective for uniform user distributions. In other words, it is not effective for distributions where the variability occurred in actual environments. This is because when the user distribution in a target area is biased, the coverage circle becomes large, resulting in much useless coverage. If the coverage circle becomes large, the overlap between the coverage circles increases and the interference increases.

In this paper, we propose a method to solve the problem of the K-means method, considering the characteristics of millimeter-waves. It is a method that introduces the smallest-circle problem to the K-means method. Here, the smallest-circle problem is a geometric problem to determine the smallest circle that can contain an asymmetric point group. Details are provided in Appendix A of this paper. The complexity of the proposed method depends on the complexity of the K-means method and the smallest-circle problem, which are $O(2IKn)$ and $O(n)$ for the former and latter methods, respectively, where $K$ is the number of clusters, $I$ is the number of iterations until convergence of the K-means method, and $n$ is the number of data points, i.e., the number of users [42]. In summary, the complexity of this UAV placement method can be given by $O((2IK + 1)n)$.

Figure 3 shows the coverage circles derived by using only the K-means method for a specific user distribution and the coverage circles obtained by applying the smallest-circle problem in addition to the K-means method. Here, the number of UAVs is 5, which means the number of clusters is 5. In Figure 3, the x and y axes represent an area of 100 m × 100 m. The points in the area represent the distribution of users and the color-coding represents clustering by the K-means method. The radius of the circle shows the coverage of each UAV and varies depending on the altitude of each UAV. The UAV is located at the center of the circle. As can be seen in Figure 3, by introducing the smallest-circle problem, the coverage circle becomes smaller. In other words, the wasted coverage is reduced, with the interference also expected to be reduced due to less overlap between the coverages.

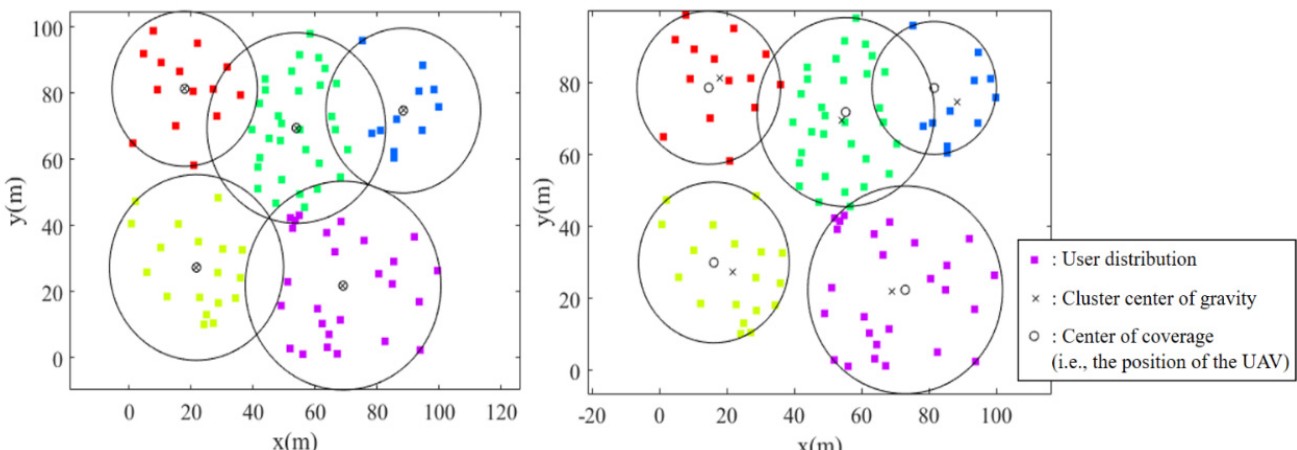

**Figure 3.** Coverage circle without (**left**) or with (**right**) adaptation to the smallest-circle problem.

### 4.2. Frequency Division for Each UAV

In this and the next section, we will introduce an interference–avoidance frequency division method after the placement of UAVs. We will also determine the association between user and UAV, i.e., which users are covered by which UAV.

As a previous study, [43] described a technique for frequency division and large bandwidth allocation for areas with increasing traffic in a terrestrial/satellite shared cell phone system. Unlike previous studies such as [43] and those for a conventional ground base stations system, the UAV network studied in this research can have a flexible cell configuration, so the coverage situation will change. Therefore, in this and the following section, we propose an algorithm for bandwidth allocation that is suitable for UAV networks.

Unlike macro base stations on the ground, the spacing between base stations in a UAV network is sparse at different times, and the overlap in coverage can be large. Therefore, it is necessary to reduce the interference power from neighboring UAVs. In this paper, we use frequency division as a method to reduce interference.

In this section, we divide the frequency for each UAV to reduce the interference between neighboring UAVs. Although completely dividing the frequency among all UAVs is the best way to reduce interference, it is not practical because it sacrifices the frequency utilization efficiency. Therefore, we consider using frequency division only for UAVs that contain users suffering from co-channel interference, i.e., UAVs whose coverages overlap with the others. However, in the construction of a UAV network, the overlapping coverage areas of these UAVs dynamically change in both time and space due to the mobility of both UAVs and users, so a clear frequency division rule is necessary. The proposed rules in this paper are shown in Algorithm 1. The final objective of this algorithm is to efficiently distribute the available bandwidth to each UAV. To achieve this, bandwidth is distributed according to the number of users, starting with the UAV with the largest number of overlapping UAVs, and then the bandwidth used by each UAV is determined. By distributing the data in such a way, the overall data rate provided can be improved.

---

**Algorithm 1:** Frequency division for each UAV

---

**Input**: Number of UAVs $K$, Horizontal position of $K$th UAV $(x_k, y_k)$, radius of $K$th UAV's coverage $r_k$, Number of users for which the $K$th UAV is providing the data rate $N_k$, Bandwidth $B$

**Output**: Bandwidth of the $k$th UAV $B_k$

   1. $\{B_{\bar{k}}\} \leftarrow \varnothing$
   2. Find $O_k$.
   3. sort($O_k$). Process 4 to 9 are performed in order from the UAV with the most overlapping UAVs.
   4. $\{B_{\bar{k}}\} \leftarrow$ UAV bandwidths that overlap with the $k$th UAV, which values have already been determined.
   5. **If** $\mathrm{card}\left(\{B_{\bar{k}}\}\right)$ = Number of UAVs that overlap the $k$th UAV
   6. $B_k = B - \sum B_{\bar{k}}$
   **7. Else**
   8. $B_k = \left(B - \sum B_{\bar{k}}\right) \times \frac{N_k}{N_k + \sum N_{\bar{k}}}$
   **9. end**

---

Here, $O_k$ is the number of UAVs overlapping the coverage of the $k$-th UAV and $\{B_{\bar{k}}\}$ represents the set of bandwidths of UAVs that overlap with the $k$-th UAV's coverage and whose bandwidths have already been determined in prior steps. In addition, $\varnothing$ represents an empty set, card() represents the number of elements in the set, and sort() performs the process of sorting in a descending manner. As an example, consider the case of $k = 3$ in Figure 4. For instance, let us assume the bandwidths of cell 1 and cell 2 have been determined, but the bandwidth of cell 6 is yet to be determined, so $\{B_{\overline{k=3}}\} = \{20\ \mathrm{M}, 30\ \mathrm{M}\}$. In this case, since the condition in Process 5 is invalid, Process 8 is applied. Moreover, $\{N_{\bar{k}}\}$ represents the set of the number of users accommodated by the UAVs that overlap with the $k$-th UAV's coverage and whose bandwidths are yet to be determined.

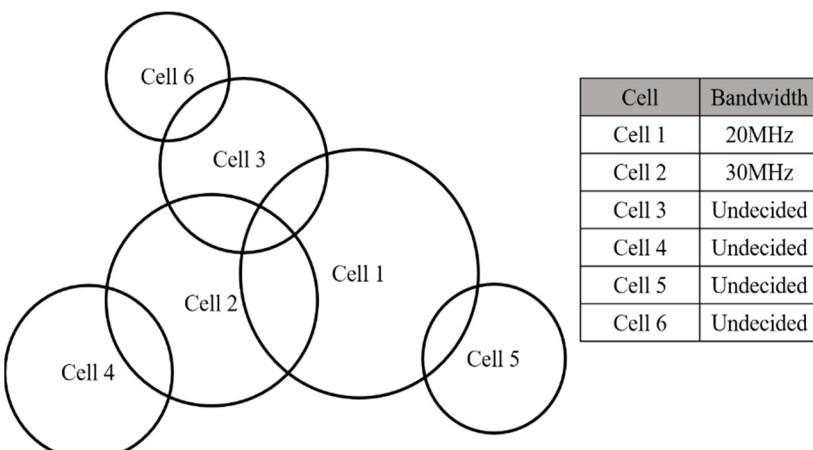

| Cell | Bandwidth |
|------|-----------|
| Cell 1 | 20MHz |
| Cell 2 | 30MHz |
| Cell 3 | Undecided |
| Cell 4 | Undecided |
| Cell 5 | Undecided |
| Cell 6 | Undecided |

**Figure 4.** Example of Algorithm 1 [19].

Process 6 of Algorithm 1 allocates all the remaining bandwidth to the $k$-th UAV when the bandwidths of all the UAVs overlapping the $k$-th UAV have already been allocated to improve the frequency utilization efficiency. In Process 8, the remaining bandwidth is divided by the number of users of the overlapping UAVs that have not yet been allocated.

### 4.3. Fractional Frequency Reuse to UAV Network

The frequency division method in the previous section narrows the bandwidth for all users, which improves throughput for users who originally suffer from high interference. On the other hand, the scheme reluctantly reduces throughput for users who originally experience low interference due to the spectrum division process. Therefore, it is necessary to investigate a novel frequency division method to only take care of the interference to users who originally suffer high interference while still guaranteeing sufficient bandwidth to original users suffering low interference, resulting in the improvement of the overall spectrum usage efficiency.

To solve this problem, we consider introducing Fractional Frequency Reuse (FFR), which has been studied in terrestrial cellular networks, to the UAV network. Figure 5a shows the outline of FFR. As shown, the area far from the base station in Cell 1, f1, which does not interfere with other base stations, is used because interference with other base stations is a concern. On the other hand, in the area around the base station where the interference with other base stations is small, all the bandwidths in use are used. Thus, in FFR, users near a base station, i.e., users who experience less interference from other base stations, are prevented from losing frequency efficiency by repeatedly using the same bandwidth as other base stations. In this way, users near the base station can be guaranteed a wider bandwidth than the case where the frequency for each base station is divided equally.

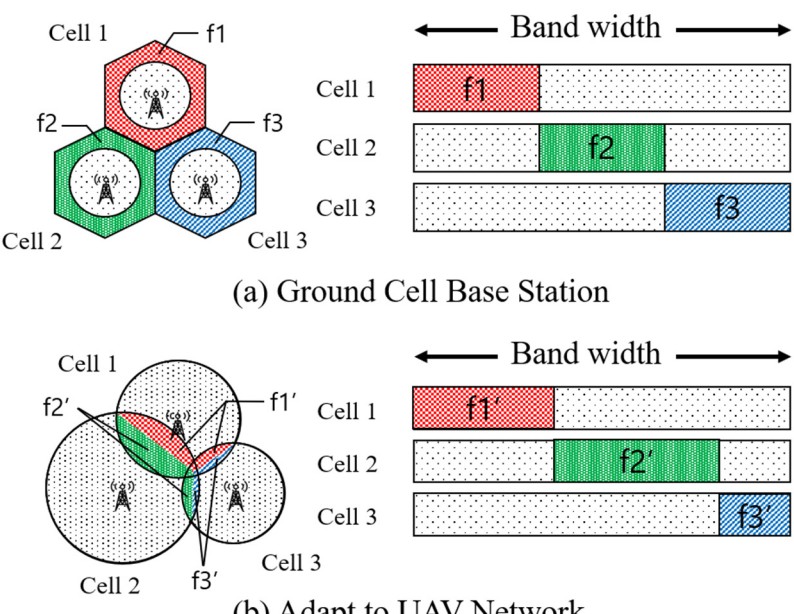

**Figure 5.** FFR overview.

We consider the application of this technique to UAV networks. In a UAV network, as described in the previous section, the number and size of overlapping coverage varies depending on the dynamic variation of the user distribution and the corresponding changes of UAV placements. Furthermore, users with little interference are not necessarily located close to the servicing base station similar to the situation on the ground. Therefore, we propose rules that can be adapted to various situations, as explained below.

Figure 5b shows a situation where three cells overlap. In this method, the bandwidth is set by the overlap of the coverage, not by the distance to the base station. As shown in Figure 5b, Cell 1 uses the entire bandwidth in areas where there is no overlap in coverage, i.e., areas with little interference. In addition, f1' is used in areas where the coverage overlaps, i.e., where the interference is high. Users in overlapping areas are served by the closest UAV. By using this rule, it is possible to clarify which UAV will provide the data even if the coverage is complicated. The details of this algorithm are shown in Algorithm 2. Algorithm 2 has much in common with Algorithm 1, and both are aimed at efficient bandwidth partitioning. However, Algorithm 2 outputs different allocation results due to the existence of Process 8 (assignment of bandwidth to coverage overlapped UEs) and Process 10.

---

**Algorithm 2:** Frequency division rate of FFR for UAV

---

**Input**: Number of UAVs $K$, Horizontal position of $K$th UAV $(x_k, y_k)$, radius of $K$th UAV's coverage $r_k$,
Number of overlapping users of the $K$th UAV providing the data rate $n_k$, Bandwidth $B$
**Output**: Bandwidth of the overlap of the $k$th UAV $B_k$, Bandwidth of the non − overlapping part of the $k$th UAV $\overline{B_k}$

1. $\{B_{\overline{k}}\} \leftarrow \varnothing$
2. Find $O_k$.
3. sort$(O_k)$. Process 4 to 10 are performed in order from the UAV with the most overlapping UAVs.
4. $\{B_{\overline{k}}\} \leftarrow$ bandwidths of the overlapping portion of the UAV that overlaps the $k$th UAV, which values have already been determined.
5. **If** card$\left(\{B_{\overline{k}}\}\right)$ = Number of UAVs that overlap the $k$th UAV
6. $B_k = B - \sum B_{\overline{k}}$
7. **Else**
8. $B_k = \left(B - \sum B_{\overline{k}}\right) \times \frac{n_k}{n_k + \sum n_{\widetilde{k}}}$
9. **End**
10. $\overline{B_k} = B - B_k$

---

In the above algorithm, $\{n_{\widetilde{k}}\}$ denotes the set of the number of users accommodated in the interference-limited area by the UAVs overlapping the coverage of the $k$-th UAV. It should be noted in Process 8 that if Algorithm 1 employs $N_{\widetilde{k}}$, i.e., the number of all accommodated users by an overlapping UAV for frequency splitting, Algorithm 2 uses only $n_{\widetilde{k}}$, i.e., the number of users in the interference-limited overlapped area for frequency splitting in only the outer area that might suffer from inter-cell interference.

### 4.4. Protocol for NFV/SDN-Based Constructions of the Proposed Networks

Based on the algorithm discussed in previous sections, we discuss the network configuration for dynamically changing the frequency assignment of the access-side UAVs. Figure 6 shows the proposed NFV/SDN-assisted UAV network architecture. It can be seen that UAV holds two interfaces: the access side for user communication and the backhaul side for communication with other UAVs or base stations. Meanwhile, the NFV/SDN controller, which controls the NFV/SDN agent in UAV, is located at the base station. The FFR partition allocation used on the access side is determined by the algorithm proposed in Algorithm 2 and controlled by SDN through cooperative coordination with UAVs and backhaul in the C-Plane (control plane). The function that determines the algorithm is the Config Manager, as shown in Figure 6. Furthermore, the Config Manager in UAV manages the CNF (Containerized Network Function) image and deploys it on the NFV platform by the NFV controller. Here, since we assume that UAV is lightweight, the computing resources of the MEC in UAV are limited. Hence, the virtualization in the MEC is not a VM (Virtual Machine) but a process bootable CNF.

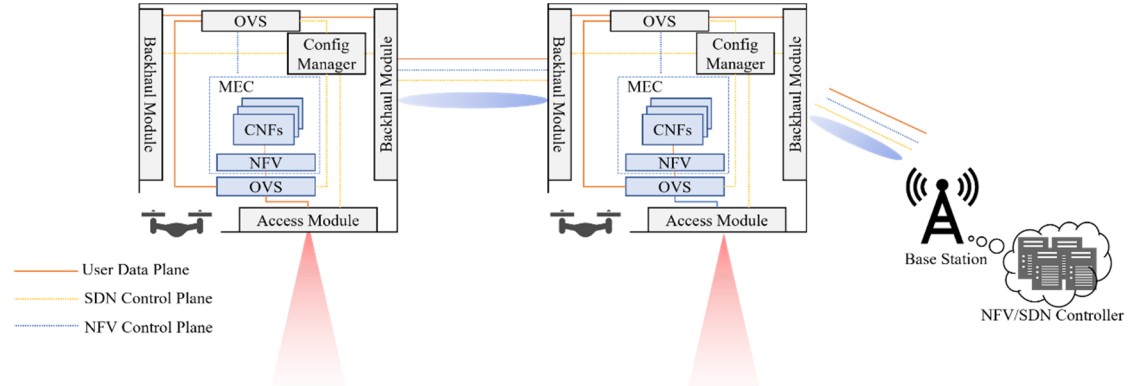

**Figure 6.** Dynamic control NFV/SDN-assisted UAV networks.

On the other hand, the SDN controller controls each UAV's access and backhaul OVS based on the network topology. In the case of CNF used by users, connecting between access and backhaul OVS with packet-through is unnecessary. Furthermore, OVS is created as an overlay network between other OVS to pass through data traffic. In other words, it is possible to construct a secure network where traffic cannot be intercepted from the outside because it is a traffic-capacitated network. Hence, access OVS is needed to communicate with CNF only in secure networks, and the SDN controller updates the flow table in OVS in UAV. Otherwise, access and backhaul OVS are needed to connect and pass through the user traffic to other UAVs or base station sides. This proposal architecture supports a dynamic control for OVS and achieves a secure platform.

## 5. Simulation Results

In this section, we compare the existing methods and the method proposed in Section 4 in terms of per-user throughput. First, we compare three UAV placement methods: the area-edge priority method (existing method in [38]), the K-means method (existing method in [41]), and the method that introduces the smallest-circle problem to the K-means method (proposed in Section 4.1). Next, as a comparison of frequency allocation, we compare three methods: a method without frequency allocation (existing method), a frequency division method for each UAV (proposed in Section 4.2), and a method introducing FFR (proposed in Section 4.3). For the UAV placement method, we use the K-means method with the smallest-circle problem (proposed in Section 4.1).

Table 1 shows the parameters used in this paper. As shown in this table, we are considering the use of a millimeter-wave with a center frequency of 28 GHz and a bandwidth of 100 MHz. The 100 MHz bandwidth of the 28 GHz band has been established as a local 5G band that can be used under license in Japan. Here, the additional loss $\eta$ is derived from the additional loss in other frequency bands described in the literature [44], and the additional loss at 28 GHz is derived by fitting. Environmental variables are variables that represent the environment of the target area. $\alpha$, $\beta$, and $\gamma$ are the ratio of buildings built to the total area of the area, the average number of buildings per unit area [/km$^2$], and the scale parameter in the distribution of building height, respectively [38]. These are used to calculate $a$ and $b$ in Equation (2) [38]. In this paper, 200 users were placed in a 100 m square area. In this research, we use a user distribution that has a densely populated area to be close to the real environment as described in Section 1. In order to do so, we use the following procedure to generate the desired user distribution. First, we randomly determine the number of hotspots from 0 to 2, and randomly set the position and size of each hotspot. Next, the ratio of the number of users in the hotspot to the overall users is randomly determined. The distribution of users outside the hotspot is then determined to be uniformly distributed over the whole area, with the distribution of users inside the hotspot determined to be uniformly distributed within each hotspot. Finally, the overall user distribution is determined by superimposing the user distributions inside and outside the hotspot [24]. In Figures 7 and 8, five UAVs were deployed for the target user distribution. In Figures 9 and 10, the number of UAVs was varied and the simulation was performed. These four numerical analyses were conducted based on 300 random generations of user distributions.

**Table 1.** Numerical parameters.

| Parameter | Value |
|---|---|
| Carrier Frequency | 28 [GHz] |
| Bandwidth | 100 [MHz] |
| EIRP * | 36 [dBm] |
| Beam Width | 45 [degrees] |
| Receive Antenna Gain | 0 [dBi] |
| Temperature | 298 [K] |
| Environmental Constant $(\alpha, \beta, \gamma)$ [38] | 0.3, 500 [km$^{-2}$], 15 [m] |
| Excessive Loss $(\eta_{LoS}, \eta_{NLoS})$ | 3.5 [dB], 47 [dB] |

* EIRP: Equivalent Isotropic Radiated Power.

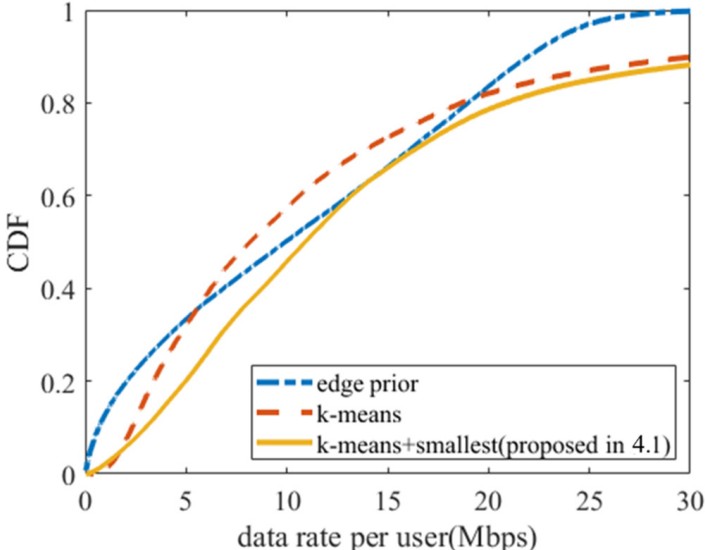

**Figure 7.** CDF of the data rate for each UAV deployment method.

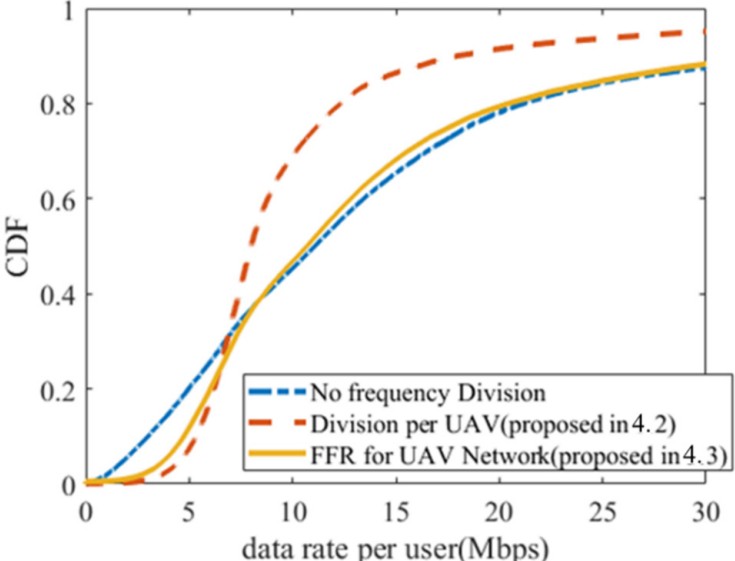

**Figure 8.** CDF of the data rate for each frequency division method.

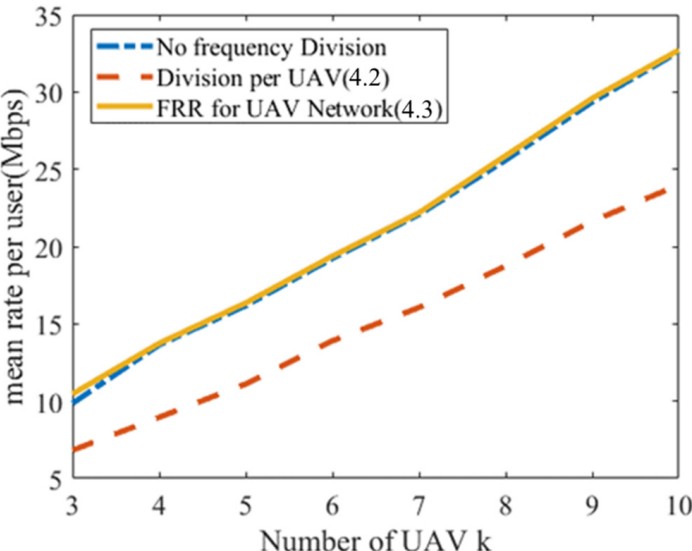

**Figure 9.** The mean data rate for each number of UAVs.

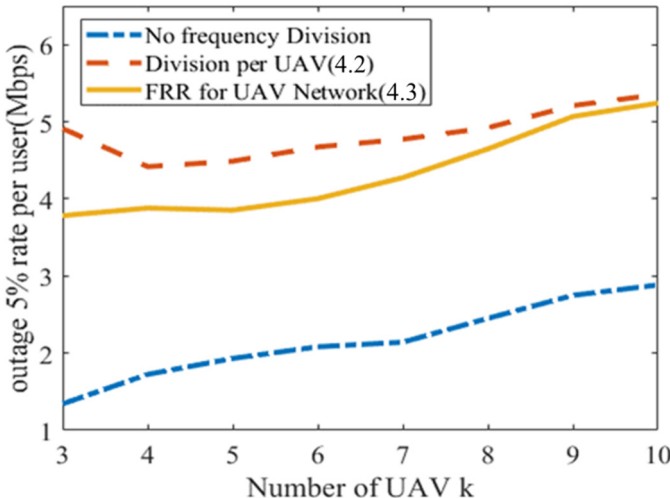

**Figure 10.** Outage 5% data rate for each number of UAVs.

Based on our simulation setups, numerical analyses were conducted using MATLAB software to compute each user's SINR from which the user data rate can be derived using equations presented in Section 3.2.2. The detailed computation process is presented in Appendix B of this paper. Figure 7 shows the CDF (cumulative distribution function) of the data rate provided to each user for each method of UAV placement, knowing that the decision of which user is associated with which UAV and the total number of users accommodated by each UAV are specified for these different placement methods. Moreover, as explained in Section 3.2.2, since this paper assumes a TDMA communication method when a UAV needs to serve multiple users, the more users that are accommodated by a UAV, the less the data rate each user has associated with this UAV experience, as the overall communication resources in the time domain need to be shared equally for each accommodated user of the cell. The edge-prior method [7] that we are comparing is an existing method and is one that ensures that the number of users accommodated by each UAV is equal, starting from users at the edge of the area. From Figure 7, we can see that by introducing the smallest-circle problem to the K-means method, the offered data rate is improved compared to the existing deployment methods. The edge-prior method results seem to be convex upward (concave downward) twice around 15 Mbps. This is because this research uses a user distribution that is not uniform. We believe that this result

can be attributed to the fact that by using this edge-prior method of accommodating a certain number of users, there are two types of users: those who are accommodated by UAVs with small coverage (and high user data rate since these UAVs fly at lower altitude close to the users) and those who are accommodated by UAVs with large coverage (and small user data rate since these UAVs fly at higher altitude to cover sparsely distributed users), yielding these two humps in performance. Figure 7 also reveals the superiority of our method (K-means + smallest circle) since the CDF curve is mostly the right-most scheme compared to the other schemes. In particular, the proposed scheme has better performance than the conventional K-means method at almost all regions of the CDF. This is due to the introduction of the smallest circle mechanism to reduce wasted coverage and also inter-cell interference. Compared to the conventional edge-prior method, the proposed method also has superior performance at almost all regions of the CDF, except for the region around 15 Mbps on the x-axis (or 65% of the CDF on the y-axis) which has almost the same performance as the edge-prior method. We believe the reason behind this phenomenon is due to the existence of UAVs that accommodate more users than these of the edge-prior method, where each UAV was restricted to accommodate the same number of users (since the algorithm in [38] was originally designed for uniform user distribution rather than the non-uniform distribution considered in this paper). Due to the existence of these "more-user" UAVs, the average user rates of these cells are slightly reduced due to the TDMA effect applied on users of the same cell. In summary, the trade-off between the reduction of the user rate due to resource sharing in the time domain and the increase of channel capacity owing to the benefit of our proposed UAV placement method explains the phenomenon at the region around 15 Mbps.

Based on the proposed deployment method, in the next step, we will check the performance of different frequency division methods to further improve the offered throughput. Figure 8 shows the CDF of the data rate provided by each user for each method of frequency division. For the UAV placement method, the proposed K-means method plus the smallest-circle problem is introduced. From the left bottom side of Figure 8, by dividing the frequency of each UAV, the offered data rate can be improved for users originally suffering high interference (with low data rates). However, the overall offered data rate is significantly reduced because the bandwidth is split even for users originally not suffering many interferences (with high data rates), as seen in the upper right side of the CDF. On the other hand, by using FFR adapted to the UAV network, we were able to increase the offered data rate for users suffering high interference while minimizing the overall loss in the offered data rate.

Figures 9 and 10 show the average data rate and outage 5% data rate when we vary the number of UAVs. As can be seen, by introducing FFR into the UAV network, regardless of the number of UAVs, we were able to increase the data rate for users at outage while reducing the loss in the average data rate. We observe that the result in Figure 9 is a linear relationship. As each UAV has an equal capability of providing the data rate, increasing the number of UAVs (or the number of base stations) yields a linear inclination of the overall system's capacity. Subsequently, the average user rate also increases linearly. The same trend is also observed in Figure 10, with the outage rate slightly increased whenever a new UAV is added to the system. This is due to the same reason for the linear inclination of the average rate in Figure 9. There is an exception for the red curve in Figure 10, which shows the performance of frequency division per UAV scheme. The outage rate first decreases when the number of UAVs changes from three to four, then follows the aforementioned trend. The authors believe that this is because when the number of UAVs is increased from three to four, the effects of interference between neighboring UAVs overwhelm the benefits of increasing UAVs.

Next, to demonstrate the benefit of the proposed mmWave-based system against the conventional microwave-band one, numerical evaluations of both systems are conducted as follows. For this purpose, the following four schemes were investigated and are shown in Figure 11.

- Microwave band/conventional K-means-based UAV placement method;
- Microwave band/proposed UAV placement method, i.e., K-means-based smallest-circle problem;
- mmWave band/conventional K-means-based UAV placement method;
- mmWave band/proposed UAV placement method, i.e., K-means-based smallest-circle problem.

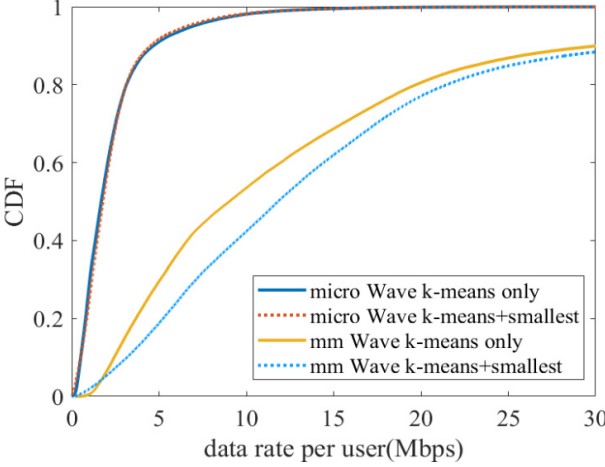

**Figure 11.** Methods' comparison with microwaves.

These methods are numerically analyzed for 200 users distributed in a 100 m × 100 m area, as in the previous studies, to obtain the offered data rate. This simulation was also based on 300 random generations of user distributions. The parameters used for the microwave-band simulation comparison are listed in Table 2.

**Table 2.** Microwave band parameters.

| Parameter | Value |
|---|---|
| Carrier Frequency | 2 [GHz] |
| Bandwidth | 20 [MHz] |
| EIRP | 36 [dBm] |
| Beam Width | 45 [degrees] |
| Receive Antenna Gain | 0 [dBi] |
| Temperature | 298 [K] |
| Environmental Constant $(\alpha, \beta, \gamma)$ [37] | 0.3, 500 $[\text{km}^{-2}]$, 15 [m] |
| Excessive Loss $(\eta_{\text{LoS}}, \eta_{\text{NLoS}})$ | 0.8 [dB], 19 [dB] |

The corresponding numerical analysis results are shown in Figure 11. As can be seen, the data rates provided by the mmWave system are generally higher than those of the microwave systems. This is due to the effect of wider bandwidth when using mmWave. When comparing the two UAV placement methods, it is found that the proposed algorithm in this paper is more efficient for the mmWave system than the conventional microwave-band system. Specifically, by using the proposed K-means-based placement method with smallest-circle problem, the offered average data rate is improved by 16.3% for the mmWave band, while it is only increased by 0.4% for the microwave band. This indicates that the proposed method is particularly effective in the higher frequency band where propagation loss is remarkable.

## 6. Conclusions

In this paper, we investigated the use of access UAV networks to provide data to disaster areas where ground base stations have been shut down due to disasters. In this paper, we reported our proposed placement method and frequency division method of

access UAVs in the UAV network and indicated that NFV/SDN technologies are key enablers for such dynamic construction of the proposed networks.

For static users, the introduction of the minimum inclusive circle problem to the K-means method improved the offered data rate by 50% compared to the conventional method without the K-means method, and by 17% compared to the conventional method using only the K-means method. As for the frequency division method, when the frequency was divided for each UAV, the average offered data rate decreased by 28% while the data rate of users with a 5% outage increased by 124%. However, by using the method of introducing FFR into the UAV network, the average offered data rate was improved by 1% while it was improved by 101% for users with a 5% outage. When the method used in this study was applied to microwaves, the performance was improved by 0.4% compared to the conventional method. In contrast, the performance of this method for millimeter-wave applications was improved by 16.3%, indicating that this method provides better results at higher frequencies.

In summary, this paper proposed an effective placement method and frequency division algorithm for millimeter-wave UAV cellular networks considering a realistic distribution of users where hotspots exist, differentiating our work from previous studies. Numerical results confirmed the effectiveness of the proposed method, in terms of not only average user rate but also outage user rate.

Investigation of the algorithm against more realistic scenarios is our ongoing work. For example, [45] dealt with a dynamic user distribution where UEs have mobility. In the future, we will also consider the placement method in realistic environments with shadowing and joint optimization together with the backhaul UAVs. In addition, although this paper presented our fundamental theoretical studies, we have been constructing a Proof-of-Concept system using actual experimental equipment and conducted real-life outdoor experiments. Preliminary experimental results had been published in [46], where we demonstrated the feasibility of millimeter-wave access of a single UAV. Via the experiment, we found out there were many natural/artificial attributes that may affect the performance of UAV networks in real life. Our ongoing work is to overcome these difficulties and demonstrate the proposed UAV system as a whole via SDN/NFV enablers.

**Author Contributions:** Conceptualization, G.K.T., M.O., and J.N.; methodology, M.O.; writing—original draft preparation, M.O., G.K.T., and J.N.; data curation, M.O.; funding acquisition, G.K.T. All authors have read and agreed to the published version of the manuscript.

**Funding:** This research received no external funding.

**Acknowledgments:** We would like to thank the anonymous reviewers for their careful reading of our manuscript and their many insightful comments and suggestions to improve the quality of the manuscript. We would also like to acknowledge the Telecommunications Advancement Foundation for its financial support to complete part of this research.

**Conflicts of Interest:** The authors declare no conflict of interest. The funders had no role in the design of the study; in the collection, analyses, or interpretation of data; in the writing of the manuscript, or in the decision to publish the results.

## Appendix A

In this section, we describe the details of the smallest-circle problem introduced in Section 4 of this paper. The algorithms are summarized in Algorithms A1 and A2.

As mentioned in this paper, the smallest-circle problem is the smallest circle that contains all points given a set of multiple points. The smallest-circle problem has been shown in the literature [36] to be computable in O(n) time but, in this study, we use the following algorithm for simplicity.

The algorithm can be divided into the following two steps:
1. Find the convex hull of the point cloud;
2. Find the minimum inclusive circle from the elements of the convex hull.

A convex hull is the smallest convex polygon that encloses a set of points. Since the smallest circle can be derived from two or three of the components of the convex hull, we first find the convex hull. There are many different algorithms for finding the convex hull. However, we believe that the convex hull of the results obtained by using those methods is identical and has no effect on the proposed method in this paper. Each algorithm is shown below.

---

**Algorithm A1:** Find the convex hull method

---

**Input**: point cloud $(x_k, y_k)$
**Output**: Convex hull of a point cloud $(Q_n)$

      1. Let $Q_1$ be the point with the minimum y-coordinate value.
      2. Let the angle with the x-axis be $\theta$ concerning the line segment from $Q_1$ to the other point group, and let $Q_2$ be the point where $\theta$ is minimized.
      3. **While** $Q_{n+1} = Q_n$
      4. Let $Q_n$ be the most recently set point.
      5. For the line segment between $Q_n$ and the rest of the points, the angle between the line segments $Q_n$ and $Q_{n+1}$ is $\theta$, and let $Q_{n+1}$ be the point where $\theta$ is minimum.
      6. **End**

---

**Algorithm A2:** Find the smallest-circle method

---

**Input**: point cloud $(x_k, y_k)$, Convex hull of a point cloud $(Q_n)$
**Output**: Information about the smallest $-$ circle (X, Y, R)

      1. While Repeat until the set circle encompasses all the point clouds.
      2. Select any three points in the vertex set $(Q_n)$ of the convex method.
      3.     If the triangle formed by connecting the three points is an acute triangle.
      4.     Let circumcenter be the center center (X,Y) and the distance from it to the farthest point be the radius (R).
      5.     **Else**
      6.     Let the midpoint of the obtuse pair of sides be the center (X,Y) and the distance to be the farthest point be the radius (R).
      7.     **End**
      8. **End**

---

**Appendix B**

In this section, we explain the process of how to derive the user data rate of our analysis results presented in Section 5. Based on our given non-uniform user distribution, users are generated randomly over the evaluating area. For each snapshot of the generated scene, the following process is conducted. First, the association (connection) between the UAV and the user is determined based on the specific UAV placement algorithms presented in Section 4.1. If frequency division (planning) is employed, then the different frequency resource division mechanisms explained in Sections 4.2 and 4.3 are introduced, depending on the extent of coverage overlap among the UAVs. Based on these simulation setups, numerical analyses were conducted using MATLAB software to compute each user's SINR from which each user's channel capacity can be derived using equations presented in Section 3.2.2. Moreover, since this paper assumes a TDMA communication method when a UAV needs to serve multiple users, the overall communication resources in the time domain need to be shared equally for each accommodated user of the cell. In other words, the computed channel capacity is divided by the total number of users associated with the target cell. This procedure is repeated until it reaches the maximum iteration of user generation; that is, 300 in this paper. All the derived data were then analyzed to compute the CDF as well as the average data rate or the outage data rate. The process for a given number of deployable UAVs is summarized in Figure A1.

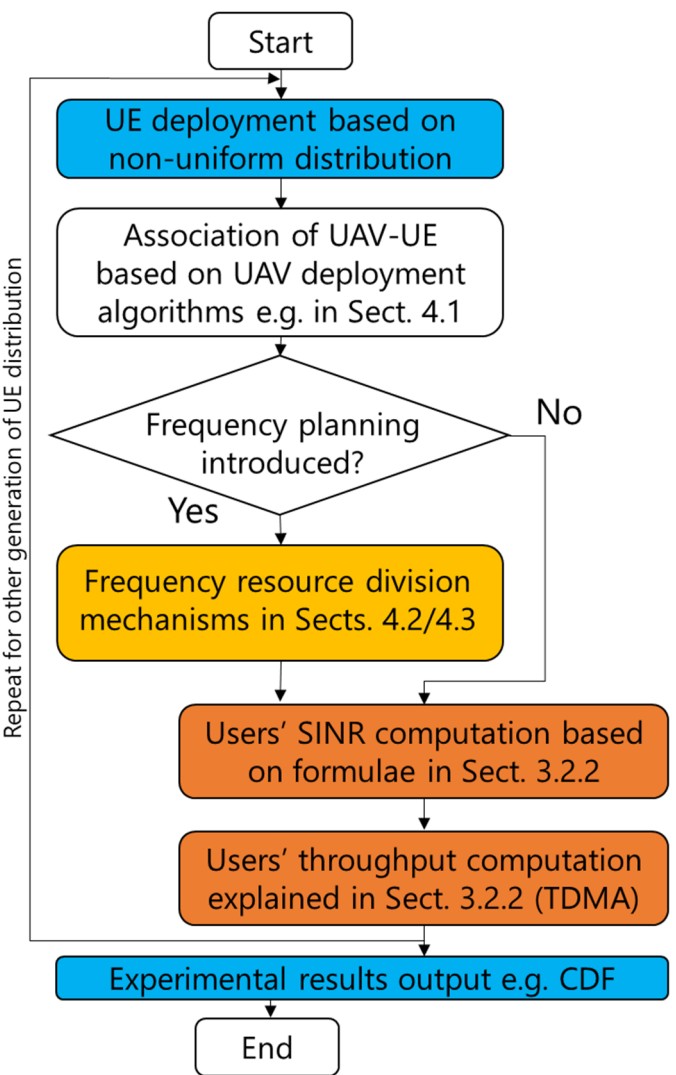

**Figure A1.** Computation process to derive numerical analysis results.

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
