# Peer review of "NFV/SDN as an Enabler for Dynamic Placement Method of mmWave Embedded UAV Access Base Stations"

_2673-8732, doi:10.3390/network2040029_

Round 1

Reviewer 1 Report

Here the paper has proposed an effective placement method and frequency division algorithm for mmWave UAV cellular networks. Good think is, it was realistic distribution of  users where the hotspots were there, this is unique. Section 5 defended the work's uniqueness like effectiveness in regards to average user rate and outage user rate. Future works are clearly defined.

Security aspects are not considered well, put a different paragraphs for that.

English has to be better so proof read it from native English speakers.

Author Response

We would like to thank the reviewer for spending his/her time to assess this submission. In the file attached, we provide a point-to-point reply to the detailed comments made by the reviewer. Please see the attachment.

Reviewer 2 Report

This paper investigates the use of UAVs (Unmanned Aerial Vehicles) networks to provide data in case of disaster when ground base stations are not working. This study is an extension of the author’s previous work. The idea of using UAV networks is interesting and the paper is well presented.  However, some issues should be resolved before publication.

The authors should mention in the introduction or conclusion that the focus of this study is access UAV placement, not backhaul UAV.  

The terms like SDN, NFV, and SINR should be expanded where first introduced in the paper like in the abstract or introduction.

In the paper, the same number is assigned to two figures i.e., 8. It reduced the reader’s understanding. Authors should check that all the figures are correctly referenced where the corresponding text is discussed.

In many places figure numbers are missing, such as on lines 432, 436, and 457 which creates difficulties in understanding the text properly.

It is written that “For this purpose, the four following schemes were investigated and shown in Fig. 13”, however, I have not found Figure 13 in the paper.

Author Response

(The authors gave the same response as above.)

Reviewer 3 Report

This paper proposed an idea of applying NFV on mmWave base stations to gain better control communication on UAV equipment. According to authors' experiment results, the offered data rate would be improved with multiple UAVs in operation. The design and development are admirable, while there are few suggestions that might be helpful improve the manuscript:

1. The introduction section has too many paragraphs. It makes content to be unorganized when reading it. Authors should consider to merge related parts to illustrate the information.

2. Authors should add more content in related work, trying comparing and discussing more survey/review papers regarding SDN and NFV, especially for signal control communication.

3. Authors may wish to explain more about the experiment results of CDF data rate in section 5. For example, how to get the throughput (i.e., 15 Mbps) result in the test? More environment information and measurement details are needed. 

4. In real practice, there are other natural/artificial attributes may make control communication of UAV to meet signal impacts. This research will be more interesting if authors could provide the evaluation/verification results of a real implementation at the outdoor. 

Author Response

(The authors gave the same response as above.)

Round 2

Reviewer 1 Report

Now all the comments are addresses very very carefully and new paragraphs and appendix are added for more clarity. So now its a good contribution.